# The archives are half-empty: an assessment of the availability of microbial community sequencing data

Stephanie D. Jurburg [1,3✉], Maximilian Konzack[1,2], Nico Eisenhauer [1,3] & Anna Heintz-Buschart [1,4]

As DNA sequencing has become more popular, the public genetic repositories where sequences are archived have experienced explosive growth. These repositories now hold invaluable collections of sequences, e.g., for microbial ecology, but whether these data are reusable has not been evaluated. We assessed the availability and state of 16S rRNA gene amplicon sequences archived in public genetic repositories (SRA, EBI, and DDJ). We screened 26,927 publications in 17 microbiology journals, identifying 2015 16S rRNA gene sequencing studies. Of these, 7.2% had not made their data public at the time of analysis. Among a subset of 635 studies sequencing the same gene region, 40.3% contained data which was not available or not reusable, and an additional 25.5% contained faults in data formatting or data labeling, creating obstacles for data reuse. Our study reveals gaps in data availability, identifies major contributors to data loss, and offers suggestions for improving data archiving practices.

[1] German Centre for Integrative Biodiversity Research (iDiv) Halle-Jena-Leipzig, Deutscher Platz 5e, 04103 Leipzig, Germany. [2] Martin Luther University Halle-Wittenberg, Halle, Germany. [3] Leipzig University, Institute of Biology, Deutscher Platz 5e, 04103 Leipzig, Germany. [4] Helmholtz Centre for Environmental Research GmbH – UFZ, Halle, Germany. ✉email: s.d.jurburg@gmail.com

Advances in microbiological research have been marked by the steady development and optimization of sequencing technologies. Where culture-dependent methods forced microbiologists to focus on a small portion of the world's microbes[1,2], high throughput sequencing methods allowed the field to bypass this limitation and indirectly observe complete microbiomes at increasingly higher resolutions. As a result, the last decade has also seen an exponential growth in the number of studies producing sequencing data, as well as in the quality of this data[3].

In particular, 16S rRNA gene sequencing, by which a section of the small ribosomal subunit's RNA gene is amplified, sequenced, and used as a tag to identify prokaryotic taxa (archaea and bacteria), has prospered during this time. 16S rRNA gene amplicon sequencing has provided microbial ecologists with census data for microbial communities similar to, or often more complete than, those obtained by macroecologists during field sampling, for example. High-throughput sequencing has supported research into the ecology of microbial communities, and a renewed interest in microbiome research. To date, individual studies have found parallels between the ecological patterns of microbiomes and those found for macroecosystems e.g., ecological scaling, species-abundance distributions, species-area relationships, and distance-decay[4,5]. However, the generalizability of these findings across ecosystems requires the systematic meta-analysis of ecological patterns across microbial communities in different environments[5-11].

Importantly, the generally uniform format of sequencing data has favored archiving practices[12,13], and the fields of genetics and molecular ecology are often cited as pioneers within ecology[14]. As microbial ecology has moved towards an increasing reliance on sequencing, the deposition of the resulting sequencing data into public genetic databases has become standard practice, and often a prerequisite for the publication in peer reviewed journals. Presently, the archiving of sequencing data is centralized in three public genetic repositories which are members of the International Nucleotide Sequence Database Collaboration (INSDC): NCBI's Sequence Read Archive (SRA), the EBI's European Nucleotide Archive (ENA), and DDJ's Sequence Read Archives (DRA)[15]. These databases are regularly synchronized and support compatible data formats, creating an opportunity for data reuse and synthesis in microbiome research. Recent meta-analyses of publicly available sequencing data have advanced the fields of medicine[9,16], microbiology[10], and microbial ecology[11].

It is expected that future advances in these areas will rely heavily on sequences which have been archived[5,7]; however, the degree to which the data which is currently archived is reusable has not been evaluated.

For archived data to serve synthesis efforts, they must be stored in findable, accessible, interoperable, and reusable formats[12]. Accordingly, INSDC databases require users to provide experiment and sample-level metadata[17] in addition to raw sequence data. In turn, these databases provide the users with stable accession numbers and the long-term storage of their data.

To evaluate how much of the currently deposited sequencing data may serve as a resource for future syntheses, we performed a comprehensive, in-depth assessment of data availability and reusability in microbial ecology. Using a combination of custom-built text parsing algorithms and manual curation, we surveyed all the literature in 17 microbiome-specific journals, selected studies which performed 16S rRNA gene amplicon sequencing and evaluated the extent to which the data were reusable. Our study shows that the lack of data deposition to appropriate repositories, improper file formatting and inconsistent labeling affected more than half of the amplicon sequencing studies surveyed.

## Results and discussion

According to an initial keyword search, we selected the 17 most popular microbial ecology-related journals, as these were more likely to have sequence-specific data deposition instructions or requirements. We surveyed all the articles published in these journals between January 2015 and March 2019 ($n = 26,927$ articles, Supplementary Table S1), as concerns over data deposition practices began to grow in 2015[14] and were soon followed by stricter standards for data availability[12]. A custom-built pattern-based text extraction algorithm followed by manual curation, we selected those studies which performed 16S rRNA gene amplicon sequencing and listed INSDC-compliant accession numbers ($n = 2015$, Supplementary Table S1; 145,203 samples).

To confirm that our parsing algorithm did not miss accession numbers in articles containing 16S rRNA gene amplicon sequencing, we randomly selected 150 articles which mentioned 16S rRNA, but for which no accession numbers were detected, for manual inspection. Of these, one contained a misspelled accession number, two had archived their sequences in unconventional repositories (Google Drive and GEO, a gene expression database, Supplementary Data 2), and 19 were identified as having

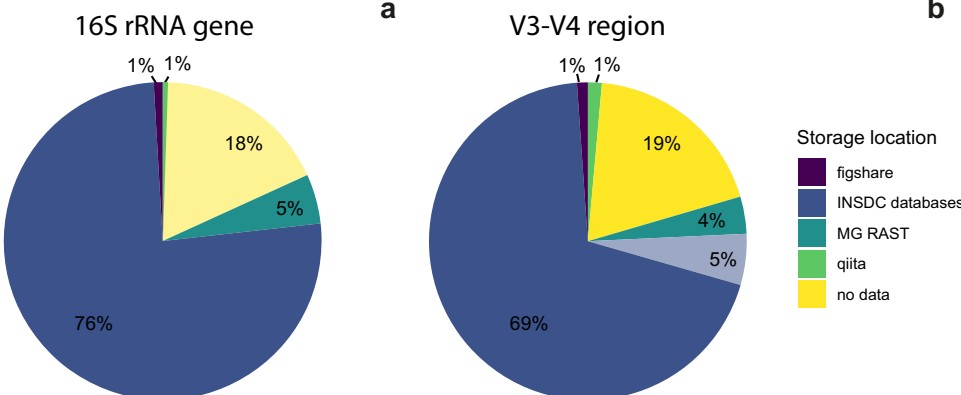

**Fig. 1 Popular locations for data storage.** Data for all studies which contained 16S rRNA amplicon sequencing (**a**), and the V3–V4 subset (**b**); $n = 2656$ and $n = 635$ studies, respectively. For the entirety of the study, studies which contained amplicon sequences but did not deposit them were inferred by manually checking 150 randomly-selected articles which did not contain INSDC accession numbers or refer to alternative databases, indicated in lighter yellow. For the V3–V4 subset, studies which contained the keywords "16S rRNA", "515", and "806" were selected. Studies for which INSDC-compliant accession numbers were reported but which did not exist on any INSDC database are shown in lighter blue.

performed 16S rRNA gene amplicon sequencing, but had not included any reference to the data. We found no cases in which accession numbers or sequence data were stored in supplementary materials. From this group, we estimate that 18% of the studies in our database ($n = 469$) performed 16S rRNA gene amplicon sequencing but did not provide access to the data (Fig. 1a). Four studies mentioned deposition data in dbGaP[18], and we could verify the existence of three of these studies. We found that an additional 6.5% of the studies had deposited their data in the Qiita[19], MG-RAST[20], and figshare databases ($n = 14$, $n = 134$, and $n = 24$ studies, respectively). Of the estimated 2,656 studies employing 16S rRNA gene amplicon sequencing, 75.9% deposited their data to an INSDC database in the period studied (Fig. 1a).

To obtain more precise estimates of the percentage of articles which deposited their data in each database, we focused on the subset of 635 studies which sequenced the V3–V4 region of the 16S rRNA gene between base pairs 515 and 806 (heretofore V3–V4 subset), a target region which has gained popularity since its development and use by the Earth Microbiome Project[21,22]. Of these, 74.5% ($n = 474$) studies listed INSDC-compliant accession numbers within the article, but of these, accession numbers from 5% of the studies ($n = 33$) were not findable on any INSDC database. Additionally, 19% ($n = 121$) did not provide an identifiable link to the data, and 6.8% of the studies deposited their data in the Qiita, MG-RAST, and figshare databases ($n = 9$, $n = 24$, $n = 7$, respectively, Fig. 1b). Two studies provided SRA submission IDs rather than accession numbers, and were also inaccessible.

The increasing popularity of microbial community sequencing was evident in our data. Over the period studied, the number of studies in the V3–V4 subset rose from 56 in 2015 to 214 in 2018 (Supplementary Fig. 1a). The proportion of publications which claimed to deposit data to INSDC databases increased slightly over time, from 33/56 in 2015 to 172/214 in 2018 ($\chi^2 = 6.6$, $p = 0.01$, Supplementary Fig. 1b), suggesting an increasing tendency towards deposition in INSDC databases. Deposition to alternative databases decreased ($\chi^2 = 14.04$, $p < 0.001$, Supplementary Fig. 1c), indicating a switch to these standardized databases but not towards making data accessible in general, as the proportion of studies which did not deposit their data was remarkably stable over time ($\chi^2 < 0.28$, $p = 0.6$, Supplementary Fig. 1d). During this period, the number of studies without publicly available data rose, from 13 in 2015, to 38 in 2018 (Supplementary Fig. 1d).

Data deposition to any public repository is preferable over no deposition at all. However, despite the advantage of using the same platform for the housing, (re-)analysis, and storage of data, non-INSDC alternatives were not designed for the long-term storage of 16S rRNA amplicon sequencing data, and thus are likely to lead to the long-term loss of information. Qiita's intended use is "the analysis and administration of multi-omics datasets" (https://qiita.ucsd.edu/). This platform is not designed for the long-term archiving of these data, and accordingly, Qiita includes software to facilitate deposition of sequences to the ENA, at which point MIMARKS requirements are enforced[17]. Similarly, MG-RAST[20] is an online platform for metagenomics analyses which also facilitates sequence deposition to appropriate databases. In contrast, figshare is a general repository which hosts most forms of research output (https://figshare.com/), but it is neither sequence-specific nor richly searchable, and does not enforce community standards.

Microbiome research spans a wide range of fields including ecology, epidemiology, medicine, biotechnology, and agricultural engineering, and is likely to become more integrative in the future[23]. Synthesis efforts to bridge knowledge gaps across environments[6] will likely rely on the ability to find data by searching databases directly, rather than resorting to a body of literature which is currently spread across the journals from various fields. To ensure future reusability, it is therefore essential that microbiome data is deposited to the appropriate INSDC databases, which also store searchable metadata and allow for automatable access to large datasets, and that current databases continue to make improvements to increase the searchability of their databases.

**Data deposition.** Due to the sensitive nature of unpublished data, INSDC databases allow users to upload their data and receive an accession number but keep the data private indefinitely[24]. This was evident in our data collection. We found that 2.2% of the studies ($n = 45$) listed incorrect accession numbers, for example placeholders (Supplementary Fig. 2b). Over the period studied, this proportion went up significantly ($\chi^2 = 9.18$, $p < 0.001$), from 1.3% in 2015 to 5.3% in 2019. Among the 2,015 articles which contained accession numbers, 7.2% ($n = 146$) of the articles had listed accession numbers correctly but had not made the sequence data public, and this proportion increased slightly over time from 5.9% in 2015 to 12.2% in 2019 ($\chi^2 = 3.9$, $p = 0.05$, Supplementary Fig. 2c), indicating that recent articles were more likely to have not made their data public at the time of manuscript publication. An additional 2.5% of the studies ($n = 51$) had not made their sequence metadata public, a trend which increased over the period studied ($\chi^2 = 14.83$, $p < 0.001$, Supplementary Fig. 2d).

**Data format.** While microbiome sequence data has been lauded for its uniform format, we found that the sequence files deposited varied quite widely in the format in which they were deposited, often rendering them unusable. Among the 441 in the V3–V4 subset for which INSDC-compliant accession numbers were available and data was public in the repository (representing 45,440 samples), we found that between 2015 and 2019, 11.8% of the studies ($n = 52$) had uploaded a single sequence file for the entirety of the project, despite analyzing more than one sample (Fig. 2b). Currently, most sequencing platforms are able to output demultiplexed data, i.e., one or more sequence file(s) per sample. However, common legacy formats consisted of one or two files for the entirety of the run as well as a mapping file, which contained the primer barcodes used to demultiplex the sequences (i.e., sequence metadata file). INSDC platforms require sequencing data to be demultiplexed prior to deposition, rendering non-demultiplexed raw data unusable due to elimination of any header information in the sequence files. Our data reflected this legacy effect: between 2015 and 2019, the proportion of studies which contained a single sequence file decreased significantly from 24.5% to 9.5% ($\chi^2 = 16.92$, $p < 0.001$, Fig. 2c). Furthermore, over this period, the proportion of studies which used Illumina platforms increased, and the proportion which used the older 454 pyrosequencing technique decreased ($\chi^2 = 10.96$, $p < 0.001$; and $\chi^2 = 10.46$, $p = 0.001$, respectively; Fig. 2c). Our findings shed light on the effect that fluxes in sequencing platform and file formats have on the scientific community's ability to access data later.

Further variability in the formatting of sequence data complicated data reuse. For example, we found that 1.6% of the studies ($n = 7$) contained sequence files which lacked standard quality scores (Supplementary Fig. 3a). During sequence processing, quality scores allow users to assess the quality of the data and to exclude sequence reads with poor quality. Therefore, sequence data lacking quality scores is not reusable. We also found that 18.1% ($n = 80$) of the studies contained putative primer sequences, but there was no significant change over time in this proportion ($\chi^2 = 2.33$, $p = 0.13$, Supplementary Fig. 3b).

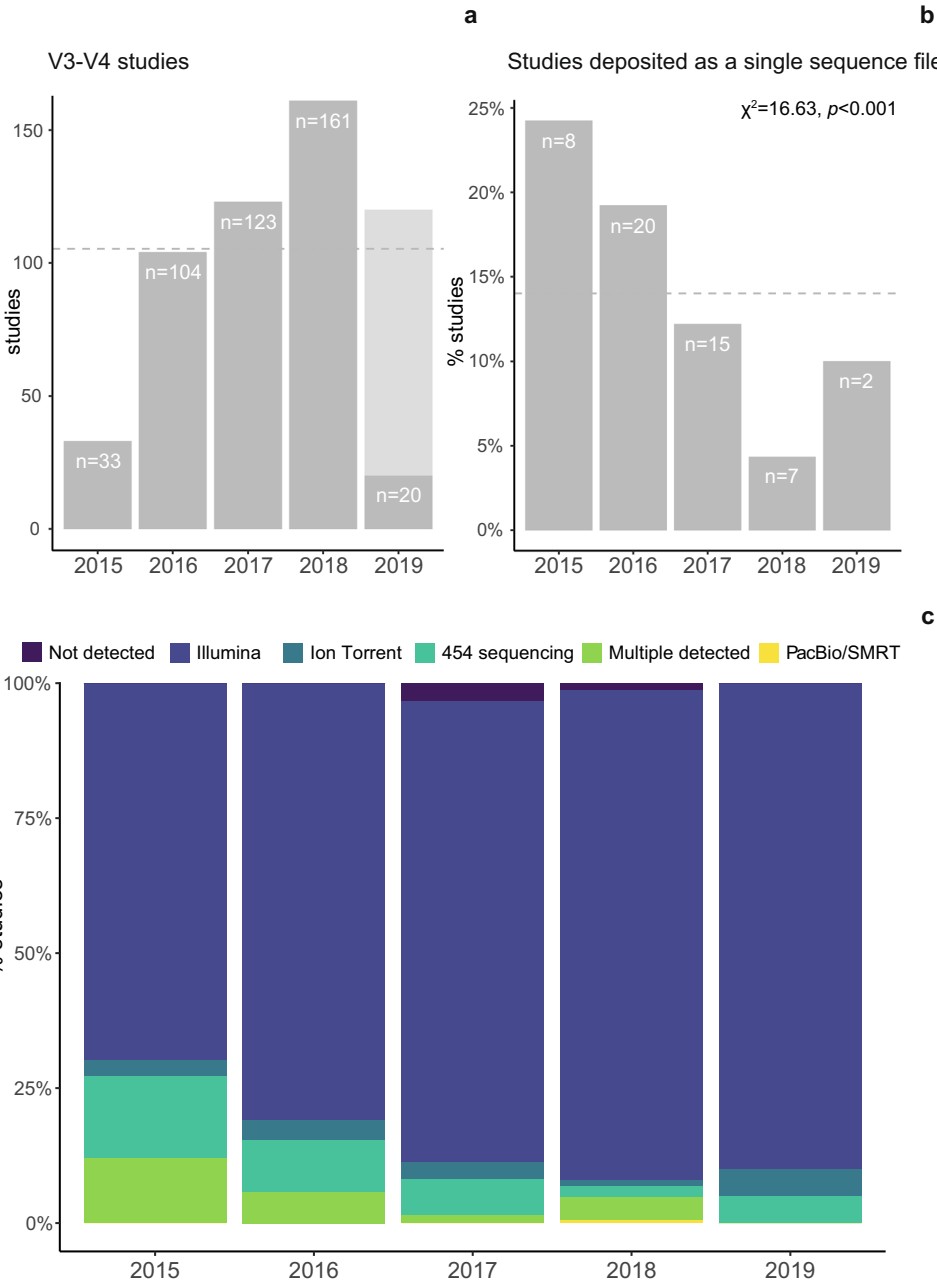

**Fig. 2 Trends in community sequencing practices over time.** The number of amplicon sequencing studies in the V3–V4 subset (**a**). The proportion of these studies which were deposited in a single sequence file a data deposition error associated with legacy sequence formats (**b**) significantly decreased over the period studied (evaluated with a Chi-squared test for trend in proportions). The proportion of each sequencing platform used across the studies changed over time (**c**). For a, the total number of articles for 2019 was estimated from the first two months of data (light gray). In **b**, the mean proportion for all years is indicated with a gray dashed line.

Primer presence is not a strong determinant of whether data is reusable, and it is advised that data is archived in the rawest format possible. However, knowledge of primer presence and primer sequence identity are essential in the proper reprocessing of the data in the future, and currently, there are no standard methodologies for including this information in the metadata. Without this information, barcode and primer sequences may be interpreted as regular data. The lack of consensus on primer presence is one example of the complexities that underlie the analysis of seemingly reusable data. Other 'hidden' obstacles include the lack of information on the formatting of quality scores, and a lack of information on the primer sequence and length. Focusing on the V3–V4 subset allowed us to collect all

possible primer sequences for this region and test for their presence; however, this is labor intensive, and forces data re-users to make inferences about the sequence formatting in their analyses, reducing the quality of research. Including extensive primer and file formatting information, as well as documentation of computational processing steps, which is automatically provided by state-of-the-art pipelines such as QIIME2[25] or Snakemake workflows[26] in the sequence metadata, may greatly facilitate data re-use.

**Data labeling**. Properly labeling sequence data and including detailed metadata is essential to data reuse[27]. Among the studies

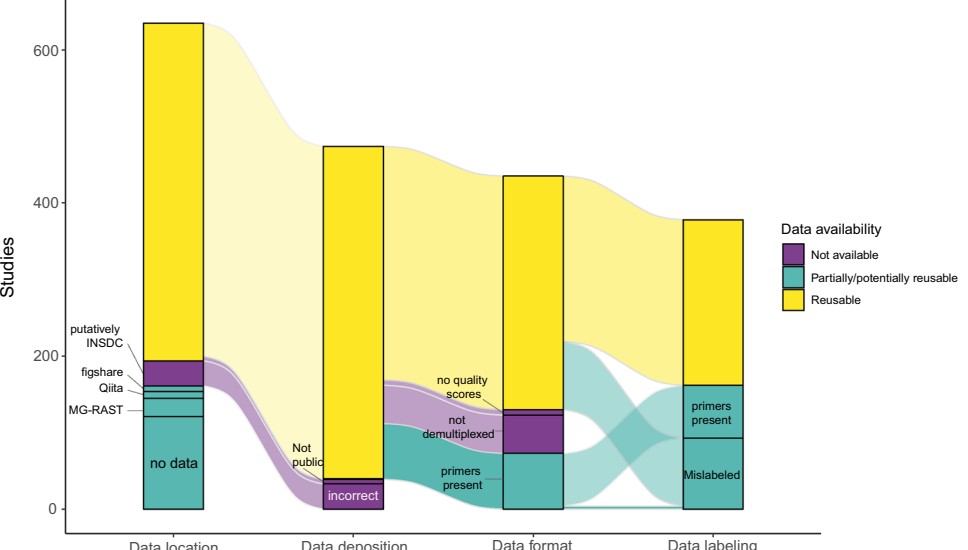

**Fig. 3 The fate of microbiome community data.** An assessment of the data location and state of the 635 studies in the V3–V4 subset. Data loss was divided into four categories: loss due to data location, errors in data deposition, errors in data formatting, and errors in data labeling. Data was categorized as 'reusable' if no faults in the above four categories were found. Data was categorized as 'partially usable' if faults in data formatting or data labeling were likely to create obstacles in data reuse (i.e., if data not findable in the database due to mislabeling). Finally, data was categorized as 'not available' if it was not publicly available on INSDC databases, or if the datasets were missing data which precluded their reusability.

in the V3–V4 subset which provided accession numbers, errors in labeling exceeded any other type of error (Fig. 3). Because our data collection contained 16S rRNA amplicon sequencing studies exclusively, we checked whether this information was included correctly in the sequence metadata. Among these studies, 12% ($n = 53$) of them had incorrectly labeled their sequences, using terms other than "Amplicon". The percentage of studies with this error varied widely, from 18.2% ($n = 6$) in 2015 to 5% ($n = 1$) in 2019, and no trends were found over time ($\chi^2 = 1.41$, $p = 0.24$, Supplementary Fig. 3c).

A defining development in the field of microbial ecology has been the advent of paired-end sequencing, by which both ends of the fragment are sequenced and later aligned in silico, resulting in a higher read accuracy or in longer read lengths[28]. Next-generation sequencers currently output forward and reverse reads in separate files. We checked whether datasets labeled as "paired" also contained files corresponding to forward and reverse reads (i.e., were labeled appropriately). This was not the case for 16.8% of the studies targeting the V3–V4 region ($n = 74$), and exhibited no temporal patterns ($\chi^2 = 2.09$, $p = 0.15$, Supplementary Fig. 3d). Much like datasets which include putative primer sequences, when data is labeled as paired-ended and only a single file per sample is available, future users must infer what the true state of the sequence data is. Upon a qualitative inspection of these datasets, we found that a common source of the error was that only the forward reads or merged reads had been deposited. This labeling error does not render the data unusable, but makes the sequencing conditions hard to understand for future users, who must reverse engineer the methods from the data format and quality information.

**Repopulating the archives.** Errors in data deposition may render entire datasets unavailable for future research, or they may greatly complicate future data reuse. To this end, we followed the 635 studies which performed amplicon sequencing of the V3–V4 segment of the 16S rRNA gene (Fig. 3). Throughout the process of archiving data, we found that 19% were not archived at

all, while 6.3% of the datasets were archived in other databases which were not designed for this task. A further 6.1% datasets were improperly deposited to sequence databases, while 11.5% and 8.9% were made partially (i.e., contained putative primer sequences) or completely unavailable (i.e., not demultiplexed) due to errors in data formatting, respectively. Finally, errors in labeling affected 14.6% of the available data.

Privacy issues, which are common in studies with human subjects, seem to have played only a minor role in choosing non-public repositories (4 studies in our dataset reported using dbGaP). One work-around for keeping microbial community data open is the removal of potentially identifying human reads. With the increasing number of more costly shotgun metagenomes, community standards for archiving either in closed databases like dbGaP or the removal step and its documentation should be formulated in the interest of re-usability without impeding privacy.

In total, only 34% of the studies identified ($n = 216$) contained fully reusable datasets, and 25.5% ($n = 162$) contained partially available datasets. A further 40.3% of studies ($n = 256$) contained data that was either not available or not reusable, severely limiting advances in synthetic microbiome research and compromising some of the fundamental principles in science[12]. An additional hurdle to data reuse is the availability of suitable metadata, and an assessment of the content and informational value of the metadata supplied for studies in the V3–V4 subset is presented in Supplementary Figs. 4–6 and 8.

Our findings show the true extent of reusability of the sequencing data which has been deposited over the past 5 years, and reveal a serious gap between the sequence data which is uploaded and that which may serve to inform future research. By identifying the main reasons for data loss (i.e., loss due to data location, errors in data deposition, errors in data formatting, and errors in data labeling), the present study provides the basis of and concrete recommendations for improved data archiving practices (Table 1). Given the plethora of pressing environmental, biotechnological, and medical challenges, preserving microbiome data is particularly relevant across fields of basic and applied research.

**Table 1 Recommendations for the future improvement of data archiving practices.**

| Studies affected | Issue | Recommendations |
|---|---|---|
| 31.4 % | Data is not readily accessible<br>• Data is not deposited<br>• Data is not deposited to INSDC-affiliated databases<br>• Accession numbers are incorrect<br>• Data is private<br>• Metadata is private | Researchers:<br>• Make deposited data available upon a manuscript's publication.<br>• Ensure accession numbers are correct in the published article.<br>• Develop community standards on removal of identifying human reads and storage of clean microbiome data.<br>Publishers:<br>• Require that the sequencing data is available upon article submission, and remind authors to make the data publicly available by the time of publication[42].<br>• Demand that datasets are deposited to the appropriate INSDC databases prior to submission in order to guarantee their long-term availability.<br>Data archives:<br>• Require that users select a date to make data public during the deposition process. |
| 23.6% | Changes in data formatting practices<br>• Data is uploaded in legacy file formats<br>• Single sequence files are uploaded for paired-end data | Researchers:<br>• Ensure that a minimum set of data is provided in order to allow for reproducibility. This includes formally collecting and depositing metadata to include experiment, sample, and sequence information; and recording protocols using modern tools[43] (i.e., protocols.io for laboratory protocols and R Notebooks or Jupyter Notebooks for bioinformatics code).<br>Data archives:<br>• Allow for the deposition of more diverse sequence file types, (i.e., allow for the deposition of sequence metadata files).<br>• Develop new standards which require the reporting of metadata on sequencing and sequence processing. Essential information such as DNA extraction, sequencing, and computational processing and data provenance should be providable via a DOI.<br>• Have a common and precise language regarding 'best practices' for data deposition (e.g., the inclusion of primers)[17].<br>• Keep publicly available changelogs of database guidelines, so that users may understand how and why data was deposited in a particular format in the past. |
| 14.6% | Mislabeling<br>• Amplicon sequences not listed as 'amplicon'<br>• Single sequence files are uploaded for paired-end data | Researchers:<br>• Become familiarized with the terms associated with sequencing and sequence formats for proper data upload[44].<br>• Proactive interaction with database holders (i.e., helpdesk) to ensure that data deposition is done correctly.<br>Publishers:<br>• Demand that the metadata tables be included during article submission for peer review.<br>Data archives:<br>• Recognize that amplicon sequencing is an increasingly interdisciplinary technique, and continue the current trend towards improved documentation and explanations. In particular, users may benefit from more precise guidelines into what constitutes informative metadata for the purposes of archiving (e.g., listing the environment as 'human' vs. 'human gut', Supplementary Fig. 4). |

## Methods

**Journal selection**. To select journals for our study, a preliminary survey of the literature was performed in February 2019 on Google Scholar with the following search query: *"bacteria" AND "515" AND "806"* to obtain a preliminary assessment of the literature employing amplicon sequencing V3–V4 region of the 16 S rRNA gene, which has been recommended and popularized by the Earth Microbiome Project[22]. Results were filtered to include only publications since 2015 and yielded ~8600 hits. The software Publish or Perish[29] was used to obtain general bibliographic information for the first 1,000 hits for the Google Scholar query (refined to *bacteria AND 515 AND 806 NOT book patent*) for each year between 2015 and 2019, yielding 4,635 results (available in Supplementary Table 1). The 17 most common journals in this list were considered the main publishers of microbial ecology data (Supplementary Table S1) and were selected for further analyses. The preprint server bioRxiv was excluded, because we only considered work that had passed the reviewing process. Similarly, we excluded journals which were not specialized in microbiology or microbiome research (i.e., *PeerJ, Nature Communications, PLOS ONE*), as specialist journals had more specific and stringent requirements for data deposition, and the authors of articles in these journals were more likely to be acquainted with microbiome data (Supplementary Data 4).

In March 2019, all articles from each of the selected journals published between January 2015 and March 2019 were downloaded as follows: the DOIs for all publications for each journal for the period studied were obtained by querying the Web of Science, using the Publish or Perish software. The concatenated list of DOIs was entered into Citavi (https://www.citavi.com) to create a bibliography and to download the corresponding articles as PDFs. Articles for which the PDFs could not be downloaded were excluded. The final set included 26,927 articles.

**PDF preprocessing, text mining, and article selection**. After renaming the entire corpus, we checked the PDF format for each file using *pdfinfo* of poppler tools (https://poppler.freedesktop.org/). We excluded invalid pdfs ($n = 12$), and applied the command *pdftotext* to extract plain text from each pdf. For each article, a corresponding searchable TEI XML file (https://tei-c.org/) was created using the GROBID v.0.5.4 command *processFulltextDocument* (https://github.com/kermitt2/grobid/). In 218 cases, GROBID was not able to generate such XML documents, and these were excluded from further analyses.

For the extraction and parsing of each TEI document, we developed and implemented a customized python package (https://github.com/komax/teitocsv). Briefly, each TEI document was parsed and searched for occurrences of general patterns including author, DOI, and journal. From the title, abstract, and main text fields (excluding references and supplementary materials), we extracted patterns indicating their relevance to this study, in particular accession numbers corresponding to INSDC-associated databases (specifically PRJ, ERP, DRP, SRP, SAME, SAMND, SAMN, ERS, ERX, DRX, SRX, DRR, SRR, ERZ, DRZ, SRZ followed by up to six digits) and references to the 16S rRNA gene, high throughput sequencing platforms, and the 16S rRNA gene region sequenced (i.e., primers). If no INSDC-compliant accession number was detected for an article, we also recorded whether alternative databases MG-RAST, figshare, or Qiita were mentioned in the text. These data were outputted as a single CSV file summarizing the findings for the entire corpus, with each accession number occupying a separate row (multiple rows per article possible) and each column capturing an aspect of pattern matching (i.e., DOI, sequencing platform). A detailed flowchart of the article selection process is included in Supplementary Fig. 7. We also searched all articles referencing the 16S rRNA gene for mention of the dbGaP database (in any

combination of capitalized letters) and verified the existence of the accessions manually. Since these data are protected, the hits were not included in the following analyses.

The accuracy of our parsing methodology was confirmed by manually inspecting 150 randomly-selected articles which mentioned 16 S but for which no accession number or alternative database was detected (Supplementary Data 2). Of these, only one article contained an accession number which was incorrectly reported (i.e., missing characters), and two had deposited their data in unconventional locations (google drive and the GEO database). Of the 150 articles inspected, none had deposited sequence files or accession numbers in the supplementary section. These articles were also used to estimate the number of articles which described 16 S rRNA amplicon sequences but did not provide an accession number for the stored data.

To assess the completeness of our data relative to all available amplicon sequencing datasets currently in existence, we conducted a Web of Science search for all articles citing the Mothur[30] or QIIME[25,31] bioinformatic tools for processing amplicon sequences on March 10, 2020, excluding all publications which were not articles or early-view articles, and had been published between 2015–2019. These workflows are the most common tools for processing amplicon sequences, hence either one is likely to be cited in articles reporting 16 S rRNA gene amplicon sequencing data. Among our 17 target journals, they were cited by 1984 articles (Supplementary Data 5).

**Data access**. To access the sequencing data, ranges of accession numbers within articles were resolved to single accessions in a two-step process: first, all potential ranges were defined based on the occurrence of multiple accession numbers with the same prefix within the same manuscript. Secondly, ranges larger than 40 accessions per study were verified manually and smaller ranges were included automatically, because all ranges between 30 and 40 were found to contain true ranges of accession numbers in a manual check. False positive accessions introduced in this step were manually removed during the final analysis.

Run-level metadata in the sequence read archive were mined for all accessions via the NCBI's Entrez Direct (EDirect) toolkit's *esearch* and *efetch* (esearch -db sra -q <ACCESSION> | efetch -format runinfo) on the 28th June 2019. We manually curated 964 articles containing accession numbers that did not yield any data, or that yielded metadata which was not labeled with the library strategy "AMPLICON" to exclude 577 articles from further analysis that did not report amplicon sequencing results of phylogenetic marker genes. Accession numbers from articles that were verified to report amplicon sequencing results but did not lead to sequence read sequencing metadata were manually confirmed on the NCBI web portal.

**V3–V4 subset**. To assess the validity of the submitted (raw) sequencing data, we focused on the accessions mined from articles which mentioned the most frequently used primer pair 515F and 806R[8,32] (Supplementary Fig. 8). Specifically, for the 515F primer, we captured any combination of the occurrence of 515F(wd) or F(wd) 515 which was separated an arbitrary number of white spaces in addition to barcoded versions of the original and the modified 515F primer. For the 806R primer, we exhausted all variations of this primer analogously. Our procedure also captures non-minimal mentions (i,e, 806Rb). For this dataset (referred to as 515F-806R-subset), 1,000 reads per submitted fastq-file were downloaded on July 2nd 2019 using NCBI's prefetch, vdb-validate and fastq-dump tools (runs specified in the library layout field as single-end sequencing: prefetch <RUN-LEVEL ACCESSION> && vdb-validate <RUN-LEVEL ACCESSION> && fastq-dump -X 1000 <RUN-LEVEL ACCESSION>; runs specified as paried-end sequencing: prefetch <RUN-LEVEL ACCESSION> && vdb-validate <RUN-LEVEL ACCESSION> && fastq-dump -X 1000 --split-3 <RUN-LEVEL ACCESSION>[33]. In studies containing only one fastq file, potential barcodes were extracted by trimming reads starting from the 515F or 806R primer sequences using cutadapt version 1.18[34]. The number of reads containing potential barcodes of 4–30 base length adjacent to the primers, and the number of different barcodes per study were assessed.

**Sequencing data evaluation and metadata access**. Sequences were searched for both primers in the degenerate form[35,36] and their reverse complements allowing for 20% mismatches and requiring 10 bases overlap. Read qualities were assessed using FastQC v0.11.3[37].

All metadata connected to the samples of the 515F-806R-substudy was accessed on August 4th 2019 via the biosample accession numbers from the run-level metadata using the NCBI's Entrez Direct (EDirect) toolkit (esearch -db sra -query <BIOSAMPLE_ID> | elink -target biosample | efetch -format docsum | xtract -pattern DocumentSummary -element Attribute@attribute_name,Attribute)[38]. The retrieved metadata was collated per study using R.

**Statistics and reproducibility**. All analyses and visualizations were performed in R 3.6.1[39]. To test for changes in the percentage of datasets which fulfilled a particular condition over time, we used a chi-squared test for trend in proportions. For the metadata analyses, we focused on the metadata supplied for the V3–V4 subset of studies. To assess the environments studied in this subset, we looked at the frequency of different environments reported in the "ScientificName" field.

To assess the informative potential of different metadata fields, we divided the fields into 'mandatory' if they were present in all datasets, and 'popular optional' if they were present in more than 25% of the studies. Note that whether a field is mandatory may change over time as INSDC deposition policies are improved. To determine the informative potential of each of these fields, we divided the number of samples in each study by the number of factor levels.

**Reporting Summary**. Further information on research design is available in the Nature Research Reporting Summary linked to this article.

## Data availability
The data that support the findings of this study are available as supplementary material and at https://github.com/drcarrot/Data_availability_study, https://doi.org/10.5281/zenodo.395330740.

## Code availability
The pattern-based text extraction algorithms are available at https://github.com/komax/teitocsv, https://doi.org/10.5281/zenodo.395331341. The python and R code used for data extraction and analysis is available at https://github.com/drcarrot/Data_availability_study, https://doi.org/10.5281/zenodo.395330740.

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

## Acknowledgements

We would like to thank H.R.P. Phillips for help with data visualization, and J. Chase, R. van Klink and S. Tem for the valuable discussions. This work was supported by the German Centre for Integrative Biodiversity Research (iDiv) Halle-Jena-Leipzig, funded by the German Research Foundation [DFG FZT 118]. The study has in part been performed using the High-Performance Computing (HPC) Cluster EVE, a joint effort of both the Helmholtz Centre for Environmental Research - UFZ and the German Centre for Integrative Biodiversity Research (iDiv) Halle-Jena-Leipzig. Open access funding provided by Projekt DEAL.

## Author contributions

S.D.J. conceived of the study, wrote the manuscript and created the figures. M.K. designed and built the pipeline to extract textual patterns from the PDFs on the entire corpus. A.H.B. performed all bioinformatics analyses of the sequence data in our collection. N.E. contributed to the organization and presentation of the results. All authors contributed substantially to subsequent revisions.

## Competing interests

The authors declare no competing interests.
