## [Peer Review File · Communications Biology]

Editorial Note: *This manuscript has been previously reviewed at another Nature Research journal. This document only contains reviewer comments and rebuttal letters for versions considered at Communications Biology.*

REVIEWERS' COMMENTS:

Reviewer #1 (Remarks to the Author):

I am happy to see that the manuscript has substantially improved compared to the previous version.

Some detailed comments:

- I agree with Referee #2 that the abstract and title should reflect clearly that this is an analysis of availability in INSDC resources. I don't think this is a 'field-wide assessment', but rather an analysis of a specific subset of data, which could be used as an example for the patterns across the entire research field.
- The authors respond to my comment regarding the focus on the 515F-806R primer set with "To clarify, our study is not limited to one primer set". However, lines 89-91 of the revised manuscript state very clearly that the V3-V4 region is defined as "V3-V4 region of the 16S rRNA gene between base pairs 515 and 806". This specific focus should be mentioned clearly in the abstract and introduction.
- I am not convinced that the focus on 'specialised journals' is well justified. I still think excluding generalist journals like Scientific Reports and PLOS ONE, but also high-impact journals (such as Nature, Science, PNAS, Nat Comm) on the assumption that authors publishing in these journals aren't "experienced handlers of microbial community data" is unjust. I will agree to disagree, I would just suggest to clearly outline this limitation & justification in the introduction + discussion.
- I must have missed it, but the response to my comment regarding the focus on just data from 2015 to early 2019 ("As now better explained in the revised manuscript, this decision was based on the assumption that deposition guidelines for sequencing data have changed over time, and guidelines have become more strict pushing towards open access, especially since 2015.") is not reflected in the revised version of the manuscript?

Reviewer #2 (Remarks to the Author):

The authors have addressed my previous comments and I have no further comments on the manuscript.

Reviewer Responses

Reviewer #1 (Remarks to the Author):

I am happy to see that the manuscript has substantially improved compared to the previous version.

We thank the reviewer for the review.

Some detailed comments:

- I agree with Referee #2 that the abstract and title should reflect clearly that this is an analysis of availability in INSDC resources. I don't think this is a 'field-wide assessment', but rather an analysis of a specific subset of data, which could be used as an example for the patterns across the entire research field.

We agree. To more accurately reflect the scope of our study, we have removed the words "field-wide" from the manuscript's title, now "The archives are half-empty: an assessment of the availability of bacterial community sequencing data" We have also explicitly mentioned the INSDC databases used in the abstract.

- The authors respond to my comment regarding the focus on the 515F-806R primer set with "To clarify, our study is not limited to one primer set". However, lines 89-91 of the revised manuscript state very clearly that the V3-V4 region is defined as "V3-V4 region of the 16S rRNA gene between base pairs 515 and 806". This specific focus should be mentioned clearly in the abstract and introduction.

We have now stated in the abstract that a portion of the calculations were focused on "a subset of 635 studies sequencing the same gene region". We must state once again that this work is not limited to a single primer set, however. The full dataset of 2,015 studies spanning all hypervariable regions of the 16S rRNA gene are included in figures 1, Figure S2, Figure S8). We focused on the V3-V4 region of the 16S rRNA gene for part of the study, because it was the most commonly sequenced region in our database, as shown in Supplementary Figure 8. And this allowed us to perform analyses which would not have been possible otherwise, particularly those related to primer detection. We clarify in text that the V3-V4 area does not comprise the entirety of the study, but is simply an area of focus: *"To obtain more precise estimates of the percentage of articles which deposited their data in each database, we focused on the subset of 635 studies which sequenced the V3-V4 region of the 16S rRNA gene between base pairs 515 and 806 (heretofore V3-V4 subset)"*

- I am not convinced that the focus on 'specialised journals' is well justified. I still think excluding generalist journals like Scientific Reports and PLOS ONE, but also high-impact journals (such as Nature, Science, PNAS, Nat Comm) on the assumption that authors publishing in these journals aren't "experienced handlers of microbial community data" is unjust. I will agree to disagree, I would just suggest to clearly outline this limitation & justification in the introduction + discussion.

We now discuss our decision to screen only specialist journals in the first sentence of the Results and Discussion section: “According to an initial keyword search, we selected the 17 most popular microbial ecology-related journals, as these were more likely to have sequence-specific data deposition instructions or requirements.” We insist, however, that this was an explicit choice, rather than a limitation of our study. We do not believe that the authors publishing in non-specialist journals are not experienced handlers of microbial community data. As we make clear in Table 1 of our manuscript, we believe that data is most available when database managers, journals, and scientists work together. As described in the previous rebuttal, guidelines for data deposition in non-specialist journal are not always as specific with regards to the deposition of sequence data (see below). Combined with the fact that microbiomes are often sequenced by scientists from a wide range of areas of expertise, the lack of clear guidelines for sequence deposition in non-specialist journals was a concern, and we selected journals accordingly.

Our decision may best be illustrated by an example. Under the ‘Web of Science assessment’ tab of the Supplementary Data file, we have included a Web of Science search detailing what percentage of the literature citing QIIME and Mothur, two popular programs for the analysis of amplicon sequences, is published in each journal for reference. You will see that the two most popular journals in this list are *Frontiers in Microbiology* and *Science of the Total Environment*. According to our selection criteria, *Frontiers* qualifies as a specialist journal, and *STOTEN* does not. While the submission process in *Frontiers* demands that an accession number (preferably to an INSDC database) be included if the manuscript presents sequencing data, *STOTEN* has much more general guidelines, and does mention the submission of sequencing data specifically.

- I must have missed it, but the response to my comment regarding the focus on just data from 2015 to early 2019 ("As now better explained in the revised manuscript, this decision was based on the assumption that deposition guidelines for sequencing data have changed over time, and guidelines have become more strict pushing towards open access, especially since 2015.") is not reflected in the revised version of the manuscript?

We apologize for the oversight. We now explain our choice of only considering studies published after 2015 in the first paragraph of the Results and Discussion section: “We surveyed all the articles published in these journals between January 2015 and March 2019 ($n= 26,927$ articles, Table S1), as concerns over data deposition practices began to grow in 2015 (14) and were soon followed by stricter standards for data availability (12).”

Reviewer #2 (Remarks to the Author):

The authors have addressed my previous comments and I have no further comments on the manuscript.

Thank you for your helpful review